# A Helium Speech Unscrambling Algorithm Based on Deep Learning

Yonghong Chen [1] and Shibing Zhang [2,*]

1   School of Information Engineering, Jiangsu College of Engineering and Technology, Nantong 226006, China; chenyh@jcet.edu.cn
2   School of Information Science and Technology, Nantong University, Nantong 226019, China
*   Correspondence: zhangshb@ntu.edu.cn

**Abstract:** Helium speech, the language spoken by divers in the deep sea who breathe a high-pressure helium–oxygen mixture, is almost unintelligible. To accurately unscramble helium speech, a neural network based on deep learning is proposed. First, an isolated helium speech corpus and a continuous helium speech corpus in a normal atmosphere are constructed, and an algorithm to automatically generate label files is proposed. Then, a convolution neural network (CNN), connectionist temporal classification (CTC) and a transformer are combined into a speech recognition network. Finally, an optimization algorithm is proposed to improve the recognition of continuous helium speech, which combines depth-wise separable convolution (DSC), a gated linear unit (GLU) and a feedforward neural network (FNN). The experimental results show that the accuracy of the algorithm, upon combining the CNN, CTC and the transformer, is 91.38%, and the optimization algorithm improves the accuracy of continuous helium speech recognition by 9.26%.

**Keywords:** helium speech; speech recognition; deep learning; corpus

## 1. Introduction

Human beings have always explored the ocean, and this exploration has impacted the fields of commerce, the military and scientific research. With the developments brought about by this exploration, about half of the world's energy and means of production are obtained from the ocean, on which human beings are becoming increasingly dependent. The initiatives of the "Maritime Silk Road" and "the Belt and Road" in the twenty-first century greatly promoted the development of marine transportation, marine industry and marine energy worldwide [1]. Therefore, deep-sea diving has been widely used in navigation, ocean development, military navigation, ocean rescue and other fields [2]. In addition, more and more people are taking up deep-sea diving.

In common diving activities, also known as "scuba diving", divers rely on gas that is contained in a cylinder and composed of about 20% oxygen and 80% nitrogen to dive. Nitrogen does not affect the human body at normal air pressure. However, as the diving depth increases, nitrogen molecules fuse with nerve cells under increasingly high pressure, causing varying degrees of anesthesia. At depths of up to 30 m, a paralyzing phenomenon known as nitrogen anesthesia can occur in divers. In addition, high-pressure oxygen might cause brain poisoning, which can seriously threaten divers' lives [3]. Research has shown that divers can avoid this physiological reaction when they breathe a mixture of helium and oxygen (helium–oxygen) instead of air at high pressure [4]. Therefore, helium–oxygen mixtures have been widely used in deep-sea saturation diving. Since the physical properties of helium–oxygen ($He\text{-}O_2$) at high pressure are very different from those of atmospheric air at normal pressure, the acoustics of speech produced at great depths are very different from that produced at normal pressure. Speech produced in a high-pressure helium–oxygen environment, which is called "helium speech", sounds like "Donald Duck"

and is difficult to understand. This could affect divers' work, and even threaten their lives if not properly handled.

It is well known that breathing gases lighter than air will cause speech distortions. However, the study of helium speech did not begin until the 1960s. In 1962, the United States Navy biologists carried out the first saturated diving experiment using helium–oxygen at normal pressure. The experiment was a biological success, but due to the low intelligibility of helium speech, it was difficult for volunteers to communicate [5]. Subsequent experiments also showed that helium speech distortion interferes with smooth communication. In the same year, Beil mixed air and helium gas in certain proportions, and then, breathed through a mask and analyzed the sound spectrum of the speech. He pointed out that the formant frequency and pitch of the speech roughly increased at a certain proportion [6].

In 1967, Sergeant first gave a report on helium speech acoustics, and determined the intelligibility of syllabic words in helium–oxygen under normal pressure [7]. In 1968, Hollien and Thompson studied the intelligibility of monosyllabic English words in a variety of gases and the relationship between pressure and intelligibility [8,9]. They pointed out that communication is impossible if depth increases. Many attempts have been made to improve the intelligibility of helium speech [10–13]. Copel first split the sound spectrum into two bandwidths, and each moved independently toward lower frequencies through a beat. In the late 1960s, there were many studies on helium speech under high pressure, and many methods of unscrambling helium speech were proposed. Among them, Fant and Sonesson first explained the formant frequency variation of speech in high-pressure gas based on the speech generation model and acoustic theory.

There are various helium speech unscrambling methods. However, they are divided into time-domain processing technology and frequency-domain processing technology, mainly from the perspective of signal processing methods. In time-domain segmentation processing, the helium speech signal is divided into several segments [14]. Each segment is stretched, and then, connected to form an enhanced helium speech signal. This can correct formant frequency distortion and even pitch distortion. Linear prediction and homomorphic filtering are also used to estimate and correct the channel impulse response in helium speech. They can achieve more accurate correction of the spectral envelope [15] or reduce the impact of noise on helium speech interpretation using autocorrelation between helium speech segments [16]. In the frequency subtraction method, the heterodyne technique is used to subtract some frequencies from the original helium speech signal spectrum to reduce the resonance peak frequency [17]. In fact, the main cause of helium speech distortion is the diver's voice cavity, rather than the helium–oxygen environment itself. Using vocoder to compress the spectrum envelope of the helium speech signal not only restores the position of the vowel formant, but also preserves the resonance structure of the helium speech [18]. Richards proposed a short-time Fourier transform enhancement algorithm based on helium speech [19]. Like the segmented processing in the time domain, this follows the segmented modification cascade method, but is processed in the Fourier transform of signal segmentation. This can arbitrarily map the spectrum envelope of helium speech without changing the pitch information of helium speech. Duncan and Jack proposed a helium speech interpretation system consisting of a residual excitation linear prediction encoder [20]. They used a joint processing method in the time and frequency domains to improve the clarity and naturalness of the decoded speech output.

As saturation diving operations deepen, existing helium speech unscrambling technology, which is based on traditional time-domain or frequency-domain signal processing technology, struggles to clearly unscramble helium speech and is unable to adapt to the dynamic changes at the depths of such diving operations. The research on helium speech has met a bottleneck problem. There have been few breakthroughs in helium speech unscrambling, and few results have been published on this topic [21–23]. Machine learning has been widely used in speech recognition in recent years [24,25]. Many companies have their own products, based on their technologies, such as Apple's Siri, and have achieved

great success [26]. Speech recognition based on machine learning opens up a new way to unscramble helium speech.

Deep learning is a complex machine learning algorithm that tries to learn the internal rules and representation levels of sample data [27,28]. It has been widely used in image processing [29,30], speech recognition [31,32], natural language processing [33,34], etc. There are three typical deep learning neural network models: convolution neural networks (CNNs) [35,36], recurrent neural networks (RNNs) and deep belief networks (DBNs). A CNN is usually used for visual frame processing and the automatic extraction of visual features [37]. They are also used for acoustic signals to process spectrograms [38] or raw waveforms [39]. The authors of [40] developed a deep learning model to predict emotion from speech. A speech enhancement method based on deep learning is analyzed in [41]. The authors of [42] proposed a speech signal processing mobile application based on deep learning, which can perform three functions: speech enhancement, model adaptation and background noise conversion. The information obtained in the learning process is very useful in the interpretation of data such as text, image and speech. Deep learning has enabled many more achievements to be made in speech and image recognition.

To date, most speech recognition technologies based on deep learning have been used to recognize normal Mandarin, and there are also some speech recognition technologies for dialects. However, there has been little work on helium speech recognition technology based on deep learning, except reference [43], due to the lack of a large corpus. However, the authors only present their initial work on isolated helium speech recognition, and not on continuous helium speech recognition.

The main contributions of this paper are as follows:

1. Chinese helium speech corpora are built. When building the corpora, we design one algorithm to automatically generate label files and one algorithm to select the continuous helium speech corpus, which reduces the scale of the training set without changing the corpus size.
2. A helium speech recognition algorithm combining a CNN, connectionist temporal classification (CTC) and a transformer model is proposed to improve the intelligibility of helium speech. Furthermore, the influence of the complexity of the algorithm and language model on the recognition rate is analyzed.
3. To improve the recognition rate of the helium speech recognition algorithm for continuous helium speech, an optimization algorithm is proposed. The algorithm combines depth-wise separable convolution (DSC), a gated linear unit (GLU) and a feedforward neural network (FNN) to improve the recognition rate of continuous helium speech.

The rest of the paper is organized as follows. In Section 2, we briefly introduce the features of helium speech and existing helium speech unscramblers. Section 3 designs speech recognition networks to unscramble helium speech. Section 4 describes the process of constructing the helium speech corpora. The experimental results are shown and discussed in Section 5. Finally, the conclusions are provided in Section 6.

## 2. Nature of Helium Speech

Speech produced under a high-pressure helium–oxygen environment has distortions, which reduce the intelligibility of the speech compared with speech under normal pressure. In fact, high-pressure helium–oxygen affects speech in many ways. Some of these are small and not obvious. However, six factors lead to distortions in helium speech: the increase in formant frequency [44], the increase in formant bandwidth [45], the attenuation of higher formant amplitudes [46], variations in pitch [47,48], high noise levels and psychological factors [49]. The formant information, including formant frequency, formant bandwidth and formant amplitude, is the main reason for the low intelligibility of helium speech [19].

At present, there are many methods that can improve the intelligibility of helium speech, and all of them can be divided into two types: time-domain processing and frequency-domain processing. Usually, time-domain processing includes a tape recorder, segment processing, digital coding, convolution processing and analytic signal rooting.

Frequency-domain processing includes frequency subtraction, a vocoder and segment processing. Time-domain processing is only suitable for moderate depths. It is simple and cheap to implement, but only corrects the formant frequency shift linearly and does not perform well in high-noise environments. Frequency-domain processing is suitable for all depths. It can compensate the nonlinear formant shift and has good robustness to high noise, but its time resolution is relatively low and its complexity is far higher.

To the best of our knowledge, neither time-domain processing nor frequency-domain processing can cope with all the distortions in helium speech. The existing unscramblers of helium speech are not satisfactory regarding dynamic changes in the depth of diving operations.

## 3. Helium Speech Recognition Algorithm

Deep learning is a specific kind of machine learning. It has been widely used in image processing, speech recognition, natural language processing, etc. Deep learning has attracted wide attention in the industry. Many Internet companies have launched products based on deep learning.

### 3.1. Helium Speech Recognition Model

To realize the adaptive unscrambling of helium regarding the depth of diving operations, we propose an intelligent helium speech unscrambler, which mainly includes preprocessing, feature extraction, an acoustic model and a language model, as shown in Figure 1.

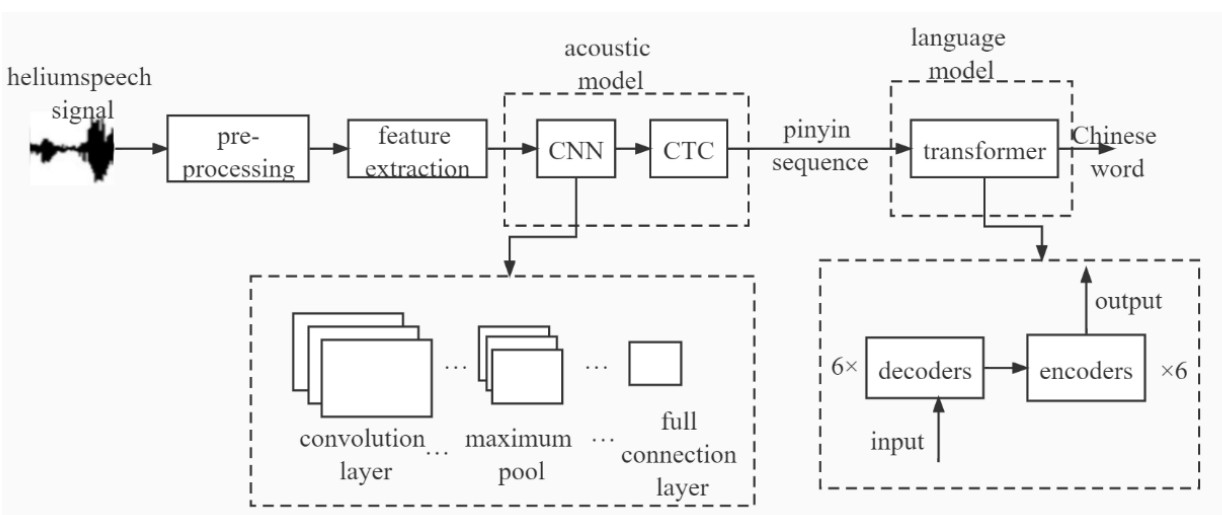

**Figure 1.** The helium speech recognition model.

Firstly, the helium speech signal is preprocessed to reduce the influence of background noise on the system, and the helium speech signal is converted from the time domain to the frequency domain. Then, appropriate representative features are extracted from the helium speech signals, which are used as the input of the acoustic model. The output consists of all possible pinyin sequences corresponding to the features and acoustic model scores. Here, all pinyin sequences refer to pinyin sequences with the same pronunciation but different text. Finally, the dictionary maps the pinyin sequence to the word sequence, and the language model estimates the probability of these sequences and generates a corresponding language model score, taking the word sequence with the highest score as the final recognition result.

### 3.2. Preprocessing

Since the process of phonation in human beings are affected by physiological factors such as the mouth and nose, the high-frequency part of human speech signals will be

attenuated to a certain extent. Therefore, to obtain a more complete speech signal, it is necessary to compensate the high-frequency part of the speech signal before the next step [50]. The implementation method is used to pass the speech signal through a high-pass filter, the system function of which can be written as:

$$H(z) = 1 - \mu z^{-1} \tag{1}$$

where $\mu$ is the pre-emphasis coefficient, which is generally between 0.94 and 0.97; 0.94 was used in this paper.

A speech signal is a constantly changing signal, which obviously does not meet the periodicity requirements. Therefore, it is not feasible to directly perform Fourier transform on a speech signal. However, the speech signal has the characteristics of "short-time stationary"; that is, the characteristics of the speech signal will maintain a stable state within a short time. During this time, the speech signal can be regarded as periodic, and Fourier transform can be performed. Therefore, we can divide a long speech signal into small segments for analysis. Each small speech signal segment is called a "frame" [51]. After the speech signal is divided into frames, windows need to be added. The purpose of adding windows is to enhance the speech waveform near the sample and weaken the rest of the speech waveform. This can reduce the spectrum leakage of the speech signal and smooth both ends of the signal. The most commonly used window functions include the rectangular window, the Hamming window and the Hanning window. The window function used in this paper is the Hamming window:

$$\omega(n) = \begin{cases} 0.54 - 0.46 cos\left[\frac{2\pi n}{N-1}\right], 0 \le n \le N-1 \\ 0, else \end{cases} \tag{2}$$

where $N$ is the length of the Hamming window.

### 3.3. Feature Extraction

Speech feature extraction is an important aspect of speech recognition. The extracted features will eventually be used as the inputs of the acoustic model. The purpose of feature extraction is to transform each frame into several representative feature values. Then, a feature vector is formed with these feature values. Mel-scale frequency cepstral coefficients (MFCC) [52] and Fbank [53] are commonly used in speech recognition. The steps used to extract these two features are described below.

Firstly, fast Fourier transform (FFT) is performed on the preprocessed speech signal to obtain the frequency spectrum.

$$X(n) = \sum_{n=0}^{N-1} x(n) e^{-j\frac{2\pi}{N}kn}, \quad (k = 0, 1, 2, \cdots, N-1) \tag{3}$$

where $N$ is the number of Fourier transform points. The spectrum is then passed through a set of Mel-scale triangular filters. There are $M$ filters in total and the center frequency of each filter is $f(m), m = 1, 2, \ldots, M$. The frequency response of the filter is as follows:

$$H_m(k) = \begin{cases} 0, \quad k < f(m-1), k \ge f(m+1) \\ \frac{2(k-f(m-1))}{(f(m+1)-f(m-1))(f(m)-f(m-1))}, \ f(m-1) \le k \le f(m) \\ \frac{2(f(m+1)-k)}{(f(m+1)-f(m-1))(f(m)-f(m-1))}, f(m) \le k \le f(m+1) \end{cases} \tag{4}$$

Then, the logarithmic energy output of each filter $s(m) = ln(\sum_{k=0}^{N-1} |X_a(k)|^2 H_m(k))$, $0 \le m \le M$ is caluculated, which are the Fbank features. Finally, MFCC features are obtained via discrete cosine transform (DCT).

$$C(n) = \sum_{m=0}^{N-1} s(m) \cos(\frac{\pi n(m-0.5)}{M}), \quad n = 1, 2, \cdots, L \tag{5}$$

where *L* represents the dimension of MFCC features.

### 3.4. Acoustic Model

The main acoustic processing networks are a CNN and CTC. A CNN is a classical deep learning network. It is a feedforward neural network, and its emergence is inspired by the biological receptive field. A convolutional neural network is essentially a mathematical model based on supervised learning.

A typical CNN network includes an input layer, a two-dimensional convolution layer (Conv2d), a pooling layer and a full connection layer [54]. The convolution layer and pooling layer can extract features from the input data, and the full connection layer can nonlinearly combine the extracted features. The number of layers determines the complexity and effectiveness of the CNN. The details of the CNN structure used in this paper are shown in Table 1.

**Table 1.** The structure of the CNN.

| Network Layer | Specific Parameters |
| --- | --- |
| Conv2d | Convolution kernel size $3 \times 3$, number 32, activation function relu |
| batch normalization | |
| Conv2d | Convolution kernel size $3 \times 3$, number 32, activation function relu |
| batch normalization | |
| Pooling layer | Maximum pooling, pooling area $2 \times 2$ |
| Conv2d | Convolution kernel size $3 \times 3$, number 64, activation function relu |
| batch normalization | |
| Conv2d | Convolution kernel size $3 \times 3$, number 64, activation function relu |
| batch normalization | |
| Pool | Maximum pooling, pooling area $2 \times 2$ |
| Conv2d | Convolution kernel size $3 \times 3$, number 128, activation function relu |
| batch normalization | |
| Conv2d | Convolution kernel size $3 \times 3$, number 128, activation function relu |
| batch normalization | |
| Pooling layer | Maximum pooling, pooling area $2 \times 2$ |
| Conv2d | Convolution kernel size $3 \times 3$, number 128, activation function relu |
| batch normalization | |
| Reshape | |
| Dropout | |

In order to align the speech segment with the corresponding text, we can formulate a rule, such as one character for ten inputs. However, different people speak at different speeds, making it difficult to formulate a rule. Manual alignment is another effective solution, but it is very time-consuming. CTC can solve the mapping problem of input and output [52]. Let *x* be a certain frame in the input sequence, *t* be time and *l* be a mapping path between the input and output. Then, the output probability is the multiplication of the transition probability of each frame on the path, which can be expressed as:

$$p(l|x) = \prod_{t=1}^{T} p(l_t|x) \tag{6}$$

Since a large amount of information is contained in speech signals, the length of feature sequences in speech is far longer than that of text sequences. CTC can effectively solve this problem.

Given an input sequence $X = [x_1, x_2, \ldots, x_T]$ and a corresponding output text sequence $Y = [y_1, y_2, \ldots, y_U]$, where *T* and *U* are the lengths of the input and output, respectively, CTC provides all possible distributions of *Y*, and then, outputs the sequence with the maximum probability $P(Y|X)$. We used loss function to evaluate the performance of CTC, which is the negative logarithmic likelihood of the output text sequence *Y*, as follows:

$$CTC(x) = -\ln p(y^*|x) \tag{7}$$

### 3.5. Language Model

The language model estimates the corresponding probability of word sequences by learning the words that appear in the text sentences, and then, generates the scores for corresponding sequences. Finally, the word sequences are sorted according to the scores generated by the acoustic model and the language model, and the word sequence with the highest score is taken as the final unscrambling result.

In our helium speech unscrambling algorithm, we use the transformer model as the language model. The transformer is mainly divided into four parts: input, encoders, decoders and output [55]. The input first enters the encoder through a multi-head attention layer. The output of the attention layer is passed to the feedforward neural network. The decoder also contains the attention layer and the feedforward layer. In addition, there is a masked multi-head attention layer [56].

The transformer adopts scaled dot-product attention to map a query and a set of key-value pairs to an output, as follows:

$$Attention(Q, K, V) = Softmax(\frac{QK^T}{\sqrt{d_m}})V \tag{8}$$

where the matrices $Q \in \mathbb{R}^{n \times d_m}$, $K \in \mathbb{R}^{m \times d_m}$ and $V \in \mathbb{R}^{m \times d_m}$ represent queries, keys and values; $n$ and $m$ denote the number of queries and keys (or values), and $d_m$ denotes the representation dimension. Multi-head attention projects $Q, K, V$ through $h$ different linear transformations and splices different attention results, as follows:

$$MultiHead(Q, K, V) = Concat(head_1, \ldots, head_h)W^O \tag{9}$$

$$head_i = Attention(QW_i^Q, KW_i^K, VW_i^V) \tag{10}$$

where $h$ is the head number, and the matrices $W^O \in \mathbb{R}^{d_m \times d_m}$ and $W_i^{Q,K,V} \in \mathbb{R}^{d_m \times d_k}$ are trainable parameters.

### 3.6. Optimization of Unscrambling Algorithm

In real scenarios, communication between divers and people on land contains many rounds of conversation, and the last sentence can provide useful information for the current sentence. In speech recognition, this information is called context sensitivity. In a model comprised of a CNN, CTC and a transformer, there are a total of 1.7 million parameters. The number of training and testing parameters and the time costs of the network are huge. To reduce the number of parameters, the proposed unscrambling algorithm should be optimized. The proposed optimization algorithm combines DSC, a GLU and an FNN to model context sensitivity, as shown in Figure 2.

DSC uses one convolution kernel for each channel, and then, splices the outputs of all convolution kernels to obtain the final output. Compared with standard convolution operation, DSC has fewer parameters and a lower cost. Additionally, the CNN in Section 3.4 uses a relu function as the activation function, and its function expression is as follows:

$$relu(x) = \begin{cases} 0 & x < 0 \\ x & x \geq 0 \end{cases} \tag{11}$$

The sparse processing forced by the relu function will reduce the effective capacity of the model, leading to the model failing to learn effective features. In the optimization algorithm, the swish activation function is used to replace relu function. The expression of the swish function is as follows:

$$swish(x) = x \cdot \frac{1}{1 + e^{-\beta x}} \tag{12}$$

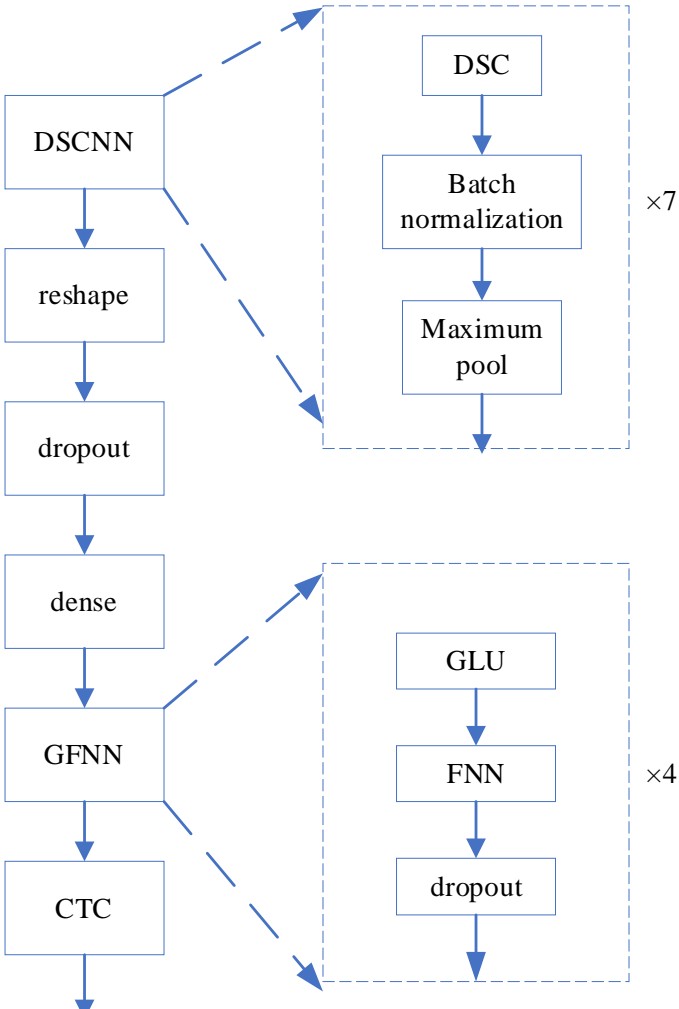

**Figure 2.** The optimization algorithm.

　　The images of the two functions are shown in Figure 3. Both functions have lower bounds but no upper bounds. Several experimental results show that the swish function has better performance than the relu function, because the swish function makes up for the shortcomings of relu and can alleviate the problem of gradient explosion.

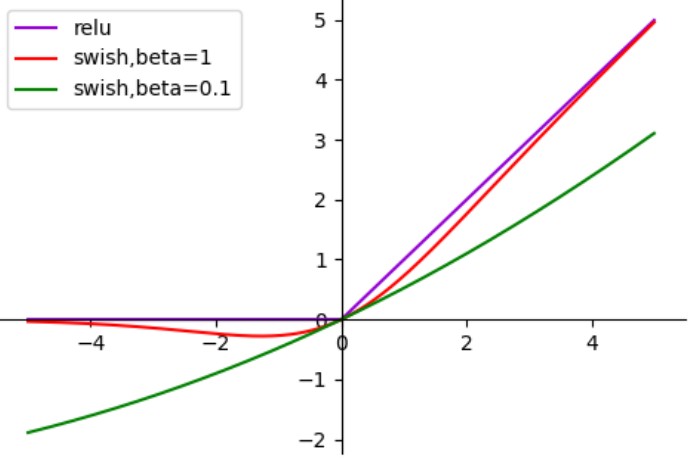

**Figure 3.** The images of relu and swish.

A GLU can be regarded as a CNN network that can process sequential data in parallel. It only has output gates, which can control the output based on the sequential information's location. The architecture of the GLU is shown in Figure 4, and the expression is as follows:

$$H = (X \times W + b) \otimes \sigma(X \times V + c) \tag{13}$$

where $X \times W + b$ and $X \times V + c$ contain the information that is passed to the next layer, $H$ represents the final output, $\sigma$ is the sigmoid activation function and $\otimes$ is a product of the matrix elements.

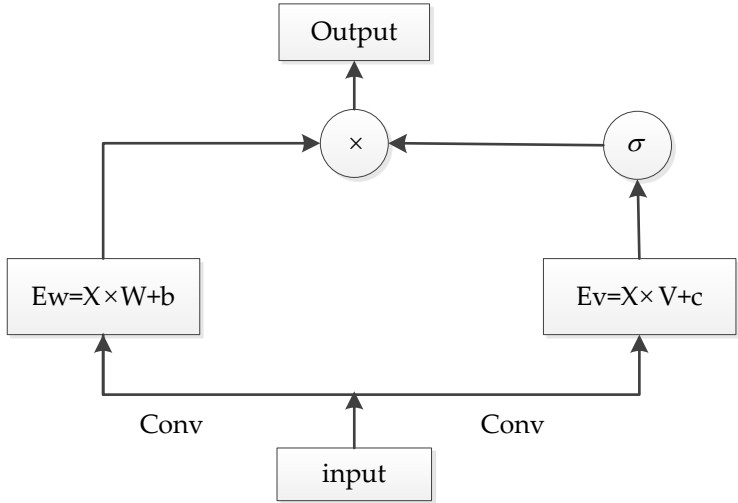

**Figure 4.** The GLU architecture.

In order to improve its ability to model context sensitivity, a feedforward neural network is added to the GLU, which can transform the output space (H) of the GLU, enhance the effective capacity of the GLU network and improve the performance of the model. The feedforward neural network is a linear change layer and consists of an activation function. The activation function also uses swish. The FNN takes the output of Figure 4, H, as the input, and can be expressed as follows:

$$\begin{aligned} \text{FNN}(H) &= \text{swish}(H \times W + b) \\ &= (H \times W + b) \cdot \frac{1}{1 + e^{-\beta(H \times W + b)}} \\ &= (H \times W + b) \cdot \text{sigmoid}[\beta(H \times W + b)] \end{aligned} \tag{14}$$

Thus, we can obtain the helium speech unscrambling algorithm, which is based on deep learning, as follows:

- Construct a helium speech corpus.
- Successively construct two corpora: an isolated Chinese helium speech corpus and a continuous Chinese helium speech corpus.
- Perform preprocessing and feature extraction (we extracted Fbank features as the input of the acoustic model).
- Train the model.
- Input the extracted features to the CNN to reduce the feature dimension, and then, input the features to CTC to obtain the pinyin sequence with the maximum probability. Input the pinyin output of the acoustic model to the transformer model to obtain the output of Chinese words.
- Test the model.
- Input the test set into the model to obtain the word error rate (WER) of the model.

Generally, traditional helium speech unscramblers require operators to judge whether the corrected helium speech is clear, which is subjective. Our proposed helium speech unscrambling method can objectively judge the effectiveness of the algorithm using the WER.

## 4. Construction of Helium Speech Corpus

Helium speech is a term used to describe the voice of divers speaking in high-pressure helium–oxygen environments. There are two necessary conditions for its formation: high pressure and a helium–oxygen environment. Moreover, deep-sea saturation diving is a highly specialized field. Divers need to acclimatize to the underwater environment in a living chamber filled with a mixture of helium–oxygen gas before working. After diving, they need to decompress; otherwise, they will suffer from the bends, which can threaten their lives. In the course of their work, a diver's daily life is very inconvenient. In sum, deep-sea saturation diving is not only time-consuming, but also dangerous, which makes helium speech a low-resource speech.

A corpus is a structured dataset of language instances that forms the heart of many natural language processing (NLP) tools. In recent years, numerous corpora of various scales have been established [57]. However, there is no corpus of helium speech available in China. In order to combine speech recognition technology with helium speech, a helium speech corpus is necessary. Due to the particularity of helium speech and the limitation of experimental conditions, a corpus of helium speech under normal pressure was constructed for preliminary study. We successively constructed two corpora: an isolated helium speech corpus and a continuous helium speech corpus. The steps that we followed to build the helium speech corpus are as follows:

(1) Selecting text content:

The first task when building a corpus is to collect corpus texts. With the popularization of the Internet in daily life, there were many text materials available for us to choose from.

For the isolated helium speech corpus, text selection was relatively simple. We selected 10 professional Chinese words in the field of communication. For the continuous helium speech corpus, we chose press releases and Mandarin exams.

(2) Collecting raw speech:

Once the text is prepared, the next task is to collect raw speech via recording. We bought dozens of tanks of 99.99% helium, and we simulated a normal-pressure helium–oxygen environment by pumping helium into balloons and inhaling it. Phones (iPhone 11) were used as recording devices, and the recording environment was a quiet office measuring 20 square meters. In order to eliminate the effect of ambient noise on text recording, we recorded the text at night when noise at its lowest. To obtain an isolated corpus, the recording was carried out by two men and one woman. Each person read an isolated word about 70 times and recorded each instance. There were 2106 recordings in total. For the continuous corpus, we made full use of the text data in the network and selected 11,890 sentences to construct a continuous corpus. These sentences were read by 14 men and 12 women and recorded. The length of all recordings was approximately 25 h. We converted the format of all audio files to WAV files and set the sampling frequency to 16 kHz.

Further, we recorded the vowel "a" in the same way and compared the formant frequency of this with that in normal environments, as shown in Figures 5 and 6. The formant information was obtained using a cepstrum [58]. The first four formant frequencies of normal speech were 941 Hz, 2603 Hz, 4231 Hz and 6064 Hz, as shown in Figure 5, and the first four formant frequencies of helium speech were 1271 Hz, 4242 Hz, 5324 Hz and 6942 Hz, as shown in Figure 6. The speech following the inhalation of helium gas showed an increase in formant frequency and variation in bandwidth compared with normal speech. This is consistent with typical features of helium speech. Therefore, it is feasible to use helium speech at atmospheric pressure instead of helium speech at high pressure for preliminary research.

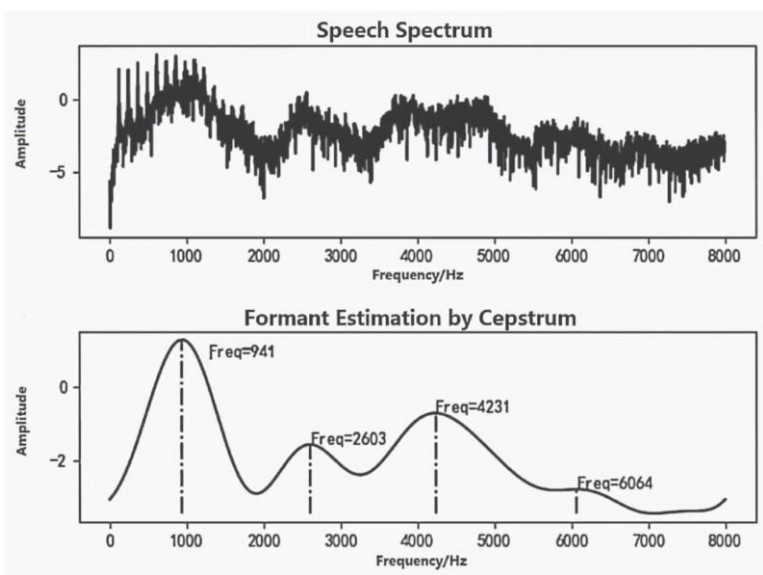

**Figure 5.** Formant information of normal speech.

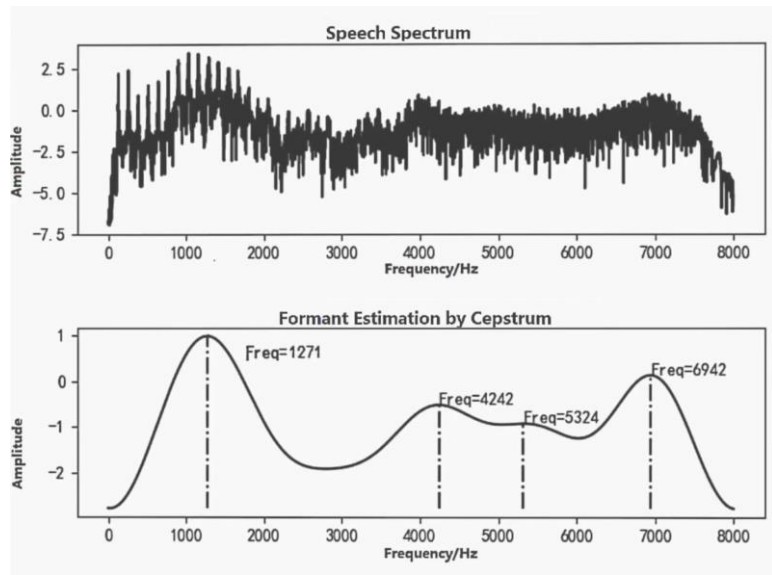

**Figure 6.** Formant information of helium speech.

(3)　Converting the file format:

The files recorded using phones were MP4 files, which needed to be converted into WAV files. To achieve this, we directly used the existing software. The sampling rate of the converted files was set to 16 KHz.

(4)　Labeling the corpus:

To save time spend on manual labeling, an algorithm was designed to automatically generate label files. For the convenience of explanation, the following parameters were introduced into the algorithm:

$Name_i$: the file name of the $i$-th file.
$Sen_i$: the text content corresponding to the $i$-th recording.
$Loc_i$: the location of the $i$-th file.
$Pin_i$: the pinyin sequence of $Sen_i$.
$Com_i$: the entire label of the $i$-th recording.
$N$: the number of sentences in the corpus.

*Lab*: the content of the label files.

The process that the algorithm used to generate label files is shown in Figure 7.

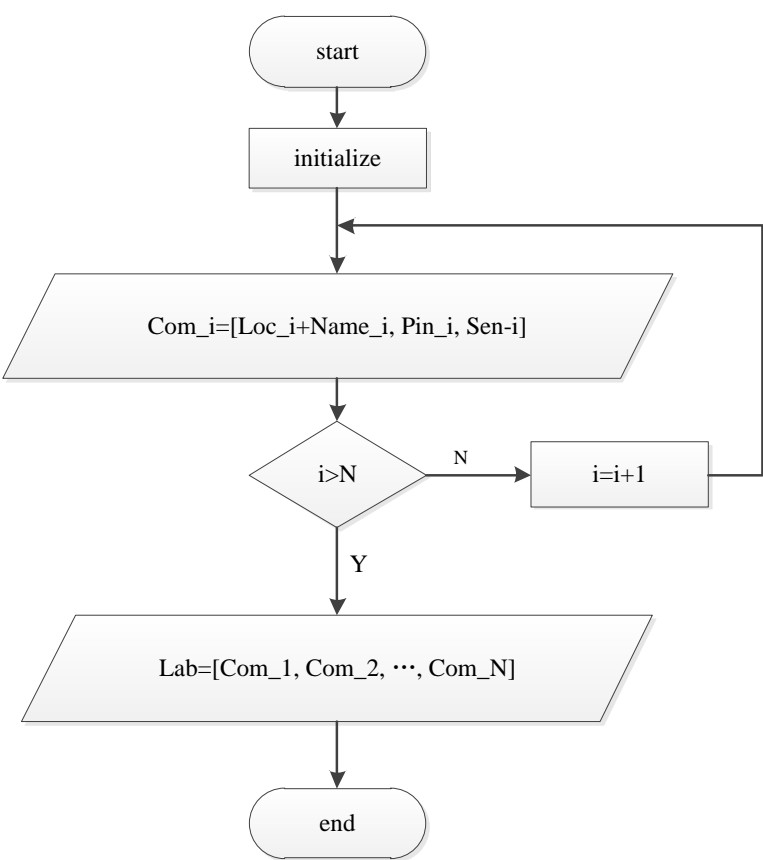

**Figure 7.** The algorithm process when generating label files.

(5)   Grouping recordings:

The available corpora generally contain training sets and test sets, and contain validation sets. In this paper, all the recordings were split into a training set, a test set and a validation set, with a ratio of 8:1:1.

## 5. Experimental Results and Analysis

### 5.1. Setup

We used the constructed helium speech corpus to evaluate the performance of our model. The CNN used in our model contained seven two-dimensional convolution layers, and the pooling layer was a maximum pool. The transformer model contained six decoders and encoders.

### 5.2. Isolated Helium Speech Recognition

We first carried out several experiments on the helium speech isolated word corpus. In order to evaluate the influence of the transformer model on the recognition accuracy, we trained and tested the isolated word model, using the CNN, CTC and the transformer (model 1), and the isolated model without a transformer (model 2). The features extracted were Fbank features. The performance of model 1 is shown with the error rate of Chinese words, while the performance of model 2 is shown with the error rate of pinyin sentences. The error rate of Chinese words is 8.62%, which is 0.32% lower than that of pinyin sentences, as shown in Table 2.

**Table 2.** WER of models with/without transformer.

|         | WER on Validation Set (%) | WER on Test Set (%) |
| ------- | ------------------------- | ------------------- |
| model 1 | 0.09                      | 8.62                |
| model 2 | 0.09                      | 8.94                |

Then, we placed the recordings of the same isolated words in one group and divided the recordings into 10 groups. We tested these 10 groups on model 1 and model 2. Table 3 shows the error rate of these ten isolated words in model 1 and model 2, respectively. The recognition accuracy of model 1 for each isolated word is higher than that of model 2.

**Table 3.** The error rate of 10 isolated words.

|         | Model 1 | Model 2 |
| ------- | ------- | ------- |
| word 1  | 3.20    | 4.50    |
| word 2  | 3.10    | 5.08    |
| word 3  | 0.84    | 0.84    |
| word 4  | 1.93    | 1.93    |
| word 5  | 10.43   | 17.53   |
| word 6  | 3.39    | 3.39    |
| word 7  | 2.41    | 2.42    |
| word 8  | 1.51    | 1.51    |
| word 9  | 3.94    | 3.95    |
| word 10 | 1.07    | 1.66    |

In Tables 2 and 3, the accuracy of the two models is shown to be good, and the model with a transformer is better. This is because the transformer model can correct the errors in the word sequence when converting the output pinyin sequence of the acoustic model into the word sequence, and then, further reduce the error rate of the algorithm.

Finally, we tested the influence of model complexity on accuracy, and analyzed the effects of the complexity of both the CNN and the transformer.

(1)    The complexity of the CNN

First, we changed the CNN complexity by changing the number of convolution layers. As mentioned above, the CNN in model 2 contained 10 convolution layers. We changed this number and trained and tested all the models. Table 4 shows the error rate of model 2 when the number of convolution layers changed. The data show that the number of convolution layers has a great impact on the accuracy. When the number of convolution layers increases, the accuracy of the model increases, but when the number of convolution layers continues to increase, the accuracy regresses. Moreover, it was found that the model training time also becomes longer with an increasing number of convolution layers. Therefore, the best result can be obtained by setting the convolution layer to seven layers.

**Table 4.** WER of model 2 with different numbers of convolution layers.

| The Number of Convolution Layers | WER (%) |
| -------------------------------- | ------- |
| 5                                | 13.82   |
| 6                                | 10.34   |
| 7                                | 8.29    |
| 8                                | 9.25    |
| 9                                | 12.42   |
| 10                               | 14.94   |

(2)    The complexity of the transformer

We changed the transformer complexity by changing the number of encoder and decoder blocks. In the transformer in model 1, there were six encoder and decoder blocks.

Then, we tested the model with the number of blocks ranging from one to six, as shown in Table 5.

**Table 5.** WER of model 1 with different numbers of blocks.

| The Number of Blocks | WER (%) |
|:---:|:---:|
| 1 | 8.46 |
| 2 | 8.46 |
| 3 | 8.46 |
| 4 | 8.46 |
| 5 | 9.11 |
| 6 | 8.29 |

As the number of blocks changed, the error rate of model 1 changed little. The highest and lowest values differed by only 0.65%. The results show that the number of blocks has little effect on the error rate of the model.

*5.3. Continuous Helium Speech Recognition*

We carried out several experiments on the continuous helium speech corpus, and the results are shown in Table 6.

**Table 6.** WER of continuous helium speech recognition model.

| | WER on Validation Set (%) | WER on Test Set (%) |
|:---:|:---:|:---:|
| model 1 | 38.31 | 42.24 |
| model 2 | 45.40 | 45.73 |

Table 6 shows the WER in the validation set and test set with/without a transformer in the model. Table 6 illustrates that the transformer model can improve the model performance. Table 6 also shows that the WER of the continuous helium speech recognition model is not ideal. The most likely reason for the high WER may be the small size of the continuous helium speech corpus. At present, most available corpora contain thousands of hours of speech, while the continuous helium speech corpus contains only about 25 h. Using a small number of data leads to model overfitting.

*5.4. The Performance of DSCNN-GFNN*

We tested the error rate of the optimization algorithm DSCNN-GFNN on the continuous helium speech corpus, as shown in Table 7.

**Table 7.** WER of different algorithms in continuous helium speech.

| | WER of Validation Set (%) | WER of Test Set (%) |
|:---:|:---:|:---:|
| ① CNN + CTC | 45.40 | 45.73 |
| ② CNN + CTC + transformer | 38.31 | 42.24 |
| ③ DSCNN + CTC | 35.94 | 37.29 |
| ④ DSCNN-GFNN + CTC | 33.82 | 36.75 |
| ⑤ DSCNN-GFNN + CTC + transformer | 29.64 | 32.98 |

Table 7 shows the WERs of different algorithms in continuous helium speech. The WER of ⑤ in the test set was 12.75%, 9.26%, 4.31% and 3.77% lower than that of ①, ②, ③ and ④. By comparing ① and ③, it can be concluded that the performance of the depth-wise separable convolution is better than that of the standard convolution, and the swish activation function is also better than the relu activation function, because both the depth-wise separable convolution and swish can improve the effective capacity of the model. By comparing ③ and ④, it can be concluded that the GLU has the ability to

model context sensitivity and eliminate useless information. The FNN can increase the GLU's capacity to store information, thus increasing the effective capacity of the GLU and improving the algorithm performance. By comparing ① and ②, and ④ and ⑤, it can be concluded that the transformer can correct errors in word sequences, thus reducing the algorithm's error rate.

Line spectrum frequency (LSF) is one of the expressions of speech signals in the frequency domain, and contains information on the formant frequency, formant bandwidth, formant amplitude and pitch [59]. We used LSF to evaluate the effectiveness of our algorithm ⑤ and compared this with the speech correction method based on the generalized regression neural network (CM-GRNN) [59] and the enhanced linear prediction coding algorithm (ELPCA) [22]. We used the letter "a", with a 30 ms frame-length and 10 ms frame-shift voice signal, as the test signal. The sampling frequency was 8000 Hz.

Figure 8 provides a comparison of the LSF performance of different algorithms. We plotted 10 LSF values. These show that the LSF of speech corrected by algorithm ⑤ is almost the same as that of raw speech. There are some errors between the raw speech and speech corrected by the CM-GRNN algorithm or the ELPCA algorithm. This suggests that algorithm ⑤ can correct the distortions of helium speech more effectively than the CM-GRNN and ELPCA algorithms.

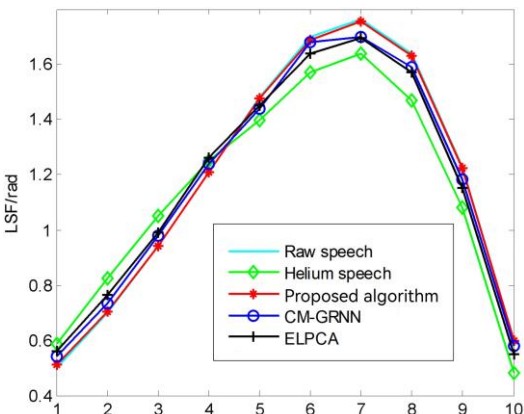

**Figure 8.** Comparison of the LSF performance of different algorithms.

## 6. Conclusions

To ensure that the unscrambler can clearly unscramble helium speech and adapt to the dynamic changes that occur in diving operations, this paper proposed a new helium speech unscrambling model based on deep learning, which combines preprocessing, feature extraction, an acoustic model and a language model. During preprocessing, we used a high-pass filter to pre-emphasize the helium speech signal and a Hamming window to reduce the framing error; during feature extraction, we used FFT and Mel-scale triangular filters to obtain the Fbank features; in the acoustic model, we used CTC to solve the alignment of the helium speech feature sequence; and in the language model, we used scaled dot-product attention to map a query and a set of key-value pairs in the transformer model. Finally, we used DSC, GLU and FNN to reduce the complexity of the model's helium speech unscrambling. Since there is no helium speech corpus available at present and it is very difficult to collect helium speech, we constructed normal-pressure helium speech corpora for preliminary studies. We trained and tested the model on the corpora using isolated words and continuous helium speech. The experimental results show that the accuracy of isolated word recognition reaches up to 91.38%, and the optimized algorithm can reduce the WER of continuous helium speech recognition by 9.26%. Further, the transformer can improve the helium speech unscrambling performance. Compared with existing unscrambling algorithms for helium speech, our proposed unscrambling algorithm can more effectively correct the distortions of helium speech.

In future work, we will collect helium speech during real diving scenarios to test the effectiveness of our model and its algorithm.

**Author Contributions:** Software, Y.C.; writing—original draft, Y.C.; writing—review and editing, S.Z. All authors have read and agreed to the published version of the manuscript.

**Funding:** This work was supported by the National Natural Science Foundation of China (No. 61871241), the Nantong Science and Technology Project (No. JC2021129) and the Natural Science and Technology Project of Jiangsu Engineering Vocational and Technical College (GYKY/2022/7).

**Institutional Review Board Statement:** Not applicable.

**Informed Consent Statement:** Not applicable.

**Data Availability Statement:** Not applicable.

**Conflicts of Interest:** The authors declare no conflict of interest.

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
