# Peer review of "A Helium Speech Unscrambling Algorithm Based on Deep Learning"

_information, doi:10.3390/info14030189_

Round 1
Reviewer 1 Report
A Helium speech Unscrambling Algorithm Based on Deep Learning
The considered manuscript is dedicated to recognition of human speech that originates with a helium-oxygen mixture, in the conditions common for sea divers.
The authors propose and implement an algorithm that relies on several existing techniques, particularly deep learning neural network. They test their approach on a "helium speech" corpus that they create, as a preliminary imitation of the real deep-divers speech.
My overall impression of the paper is positive: * the authors address an interesting and practical, though narrow, topic; * they provide a mostly comprehensive overview of the algorithm; * they create a very specific speech corpus.
So, I believe that the manuscript could be accepted after the authors fix several disadvantages, which mostly are minor. My detailed comments on them are below.
== State-of-the-art review and Evaluation ==
In the Introduction and some other sections of the paper, the authors provide an overview of several related issues: scuba diving gas, human physiology, speech distortion, etc. However, an overview of deep learning methods' application in the field is largely missing. For instance, the authors mention that "in the past decades, the research on helium speech has met a bottleneck problem" - but what kind of bottleneck and what exactly is the problem?
Another example is 120 "The existing unscramblers of helium speech are not satisfactory." - still, these should be named and considered in the SotA review.
As for the evaluation part: although I could agree with the authors' statement "At present, there are few experimental results published in helium speech recognition for comparison." - it does not mean there is no baseline for comparison whatsoever. The authors could have at least compared their results with the ones obtained for other low-resource languages.
== The novelty of the algorithm ==
This is related to my previous comment. Throughout Section 3, it is hard to grasp what is proposed by the authors and what is the background of existing approaches that they describe. Since the algorithm is the main contribution, this absolutely needs to be improved. In particular, in 3.8, the novelty of the algorithm (and its parts) should be made more clear. Also, in some cases parts of the approach miss a comprehensive justification. For instance, why exactly this CNN structure was proposed (Table 1)?
== English and writing style ==
Although generally the paper is written comprehensively and is easy to read, sometimes the authors use expressions that are uncommon or even just wrong English-wise or stylistically. These need to be corrected. Some examples are: * "It would affect divers’ work, even threaten their lives if not handle." -> "... if not handled properly?" * "2. Natures of helium speech" - Nature should not be plural * "Use phones as recording devices and the recording environment is a quiet office." -> "Phones were used..."?
* Table 3 - why has the order of the models changed (model 2, model 1)? * "Data Availability Statement: Not applicable, the study did not report any data." - how come? Particularly, given that the authors list the speech corpus among the contributions of the work?
Reviewer 2 Report
The topic is important, interdisciplinary and practical. The text is informative and written in proper English. Figures and equations are generally clear and understandable, they seem plausible and free of error.
Suggestions and comments:
1) In the Title, the word [Speech] should be written with a Capital S.
2) Several minor editorial and formatting issues are present, e.g., additional (unnecessary) space signs between subsequent words, etc.
3) Figure 3 has too low resolution and/or inappropriate file format, it looks blurred and the X and Y axes are not labeled.
4) Authors should point out how the speech corpora was recorded, including: type of microphone(s) used, bitrate, sampling rate/frequency, file format, mode (mono/stereo), etc. What was the covered frequency range (20 to 20000 Hz)? What about the room capacity/volume and its prevention from surrounding noise, etc.? How long did a single session take? Did all participants read the same set of sentences or just a single word? How long did a single audio file last and where all of them of the same duration? How many did they include? Additional comments are necessary.
5) Did the Authors really utilize only a smartphone with its build-in microphone or what?
6) Did the Authors follow any EBU/ITU, etc., recommendation when gathering their dataset?
7) The Chinese language is a general term – what kind of dialect did it include? Additionally, what about the background of participating speakers/lectors? What about their language skills, etc.?
8) Did the resulting dataset include all 100% of recorded signal samples? Additional comments seem necessary.
9) Figure 5 is too small in size, with too small fonts, making it hard to read and interpret.
10) Except the transition from MP4 to WAV, were there any other alterations made to the audio files?
11) The Conclusions section is too short and not convincing at all, it needs to be redone. Provide a summary of your findings, give feedback to the potential reader. Do write about future study directions and open issues.
This paper lacks serious and important information, yet it has a potential to be a good one. Authors are therefore advised to carefully acquaint with provided comments and suggestions, and make necessary extensions and corrections. To sum up, major revisions are necessary.
Reviewer 3 Report
A Helium speech Unscrambling Algorithm Based on Deep Learning
The authors are proposing a neural network deep learning algorithm to accurately unscramble helium speech voice. In accomplishing the task, an isolated and continuous Chinese helium speech corpus in a normal atmosphere was constructed and an algorithm to automatically generate label files was further proposed. The research work is interesting and has useful applications in the problem of helium speech voice communication in supporting the life and work of divers in the deep sea. However, I would like the authors to address the following issues before the manuscript can be recommended for publication.
1. There are grammatical errors in the text. For example, “…and other fields has…” should be replaced by “…and other fields have…”; “… has distortions which reduces…” by “… has distortions which reduce…”; “… an algorithm is …” by “… an algorithm was …” and the small letter “s” appearing in word “speech” in the title of the manuscript should be capitalized.
2. The phrase “… and transformer is over 91%” in the abstract should be replaced by the exact accuracy score.
3. The existing algorithms for helium speech unscrambles were not discussed as claimed in the summary of section 1. Section 2 misses out on a critical discussion of the extant algorithms for recognizing helium speech.
4. The authors should be specific about why the existing algorithms are not suitable. They did not compare their results with the results computed by the existing algorithms or provided strong literature evidence to conclude that existing algorithms are not appropriate. The current section 2 should be expanded to expose the prime contributions of the study reported and the specific issues addressed in the manuscript.
5. Section 3 should be restructured to ensure that the description of the proposed algorithm complies with Figure 1. It presents four main components of preprocessing, feature extraction, acoustic model, and language model that should form the basis for describing the proposed algorithm. It is unclear why sections 3.4, 3.5, 3.6, and 3.7 cannot be subsumed in the discussion of these four components.
6. Besides the term model, the title of section 3.1 is not different from that of section 3, so the subsection should be deleted. The content of section 3.8 is insufficient to constitute a separate section. I suggest that it be subsumed in section 3 to summarize the proposed algorithm before a detailed discussion.
7. Any reason for choosing a Hamming window at the preprocessing phase to enhance waveform instead of other available options?
8. CNN should be used instead of a convolutional neural network after it has been mentioned in section 3.4 to be consistent with the use of the acronym.
Reviewer 4 Report
The research paper addresses a very prevalent issue related to the development of saturation diving, which has enabled man to work at vast depths and for long periods of time in the water.
This advancement is due, in part, to the substitution of helium for nitrogen in breathing gas mixes.
However, the use of HeO 2 breathing mixes at high ambient pressures has posed challenges with verbal communication; as a result, electronic devices to assist diver communication have been developed.
This paper however seems like an enhanced variant of the original paper by prof. Zhang (also one of the authors): https://doi.org/10.1109/WCSP52459.2021.9613483
I fail to find significant changes in the approach which is my main concern. Sure the experiments are enhanced in this submission, hoever the difference in the methodology is not and requires a significant rewrite. Even some of the pictures are the same.
I will eagerly await for the resubmission of this work (and I strongly encourage to enhance this paper and resubmit) as its of interest to every speech signal analysist and could potentially have some use in the ai driven marine applications.
Round 2
Reviewer 2 Report
Authors have revised their manuscript, however not all suggestions and remarks have been addressed properly. In future papers, remember to always justify your actions – either make or do not make any changes, but give arguments on why did you chose to do so.
This paper may be viewed as a preliminary study, due to the data gathering part, and the way of handling and processing data, etc. Yet, it is a good starting point and source of inspiration for other Authors.
I encourage the Authors to continue their studies – think about using other devices, both low class and high class ones, coming from different manufacturers, and running on various operating systems. Next, consider performing additional subjective and objective quality evaluations and user expectations studies. The potential, in case of this topic, is very wide.
This paper still has some minor editorial and formatting issues, as well as English grammar and style imperfections. However, they can be overcome at a later stage, during MDPI’s proofreading. I leave the final decision to the Editor, whether the manuscript should undergo a second minor revision.
Reviewer 4 Report
Thank you for clarifying.
A more extensive architecture, performance analysis, and robustness investigation would enrich the article.
I propose include these to help distinguish the original conference materials.
Please replace generic block diagrams with UML-based use case diagrams and activity diagrams (logical code sequence) that describe the implementation.
Instructions (if necessary) may be found here:
https://www.tutorialspoint.com/uml/uml use case diagram.htm
https://www.tutorialspoint.com/uml/uml activity diagram.htm
Finally, I would consider hiring a native English speaker to repair the text's problems.
Round 3
Reviewer 4 Report
I regret writing this, but I cannot suggest accepting this work because just the wording and grammar was changed, which is insufficient to distinguish it from previously published variants in conference proceedings.
